# A Novel Evidence Combination Method Based on Improved Pignistic Probability

**DOI:** 10.3390/e25060948

**Published:** 2023-06-16

**Authors:** Xin Shi, Fei Liang, Pengjie Qin, Liang Yu, Gaojie He

**Affiliations:** School of Automation, Chongqing University, Chongqing 400044, China; 18326097499@163.com (F.L.); qinpengjie1215@163.com (P.Q.); 18226844807@163.com (L.Y.); tiantianneedlove@163.com (G.H.)

**Keywords:** DS evidence theory, pignistic probability function, information fusion

## Abstract

Evidence theory is widely used to deal with the fusion of uncertain information, but the fusion of conflicting evidence remains an open question. To solve the problem of conflicting evidence fusion in single target recognition, we proposed a novel evidence combination method based on an improved pignistic probability function. Firstly, the improved pignistic probability function could redistribute the probability of multi-subset proposition according to the weight of single subset propositions in a basic probability assignment (BPA), which reduces the computational complexity and information loss in the conversion process. The combination of the Manhattan distance and evidence angle measurements is proposed to extract evidence certainty and obtain mutual support information between each piece of evidence; then, entropy is used to calculate the uncertainty of the evidence and the weighted average method is used to correct and update the original evidence. Finally, the Dempster combination rule is used to fuse the updated evidence. Verified by the analysis results of single-subset proposition and multi-subset proposition highly conflicting evidence examples, compared to the Jousselme distance method, the Lance distance and reliability entropy combination method, and the Jousselme distance and uncertainty measure combination method, our approach achieved better convergence and the average accuracy was improved by 0.51% and 2.43%.

## 1. Introduction

As an uncertain reasoning method, evidence theory [1,2] needs weaker conditions than Bayesian probability theory, but it possesses the ability to express “uncertainty” and “ignorance” directly. The primary data required in evidence theory are more intuitive and easy to obtain than in probability reasoning theory. One can quickly integrate the knowledge and data from different experts or data sources to describe the uncertainty flexibly. It has been widely used in supplier selection [3,4], target recognition [5,6], decision making [7,8], reliability analysis [9,10,11,12,13], optimization in uncertain environments [14,15,16], etc. However, in the application of DS evidence theory, evidence fusion plays a crucial role due to the unreliability of the evidence source. The Dempster fusion rule was established based on the multiplication principle. In cases where there is conflicting evidence in evidence theory, the Dempster combination rule is used to explain counterintuitive outcomes, which can lead to what is known as the “Zadeh paradox”.

To address the issue of conflicting evidence fusion, scholars have presented many outstanding studies. When there are conflicts in original evidence, traditional DS theory cannot be applied and it needs to be improved. In recent years, a large number of scholars and experts have improved evidence theory from two aspects. The first is to improve the fusion rules of DS evidence theory. Sun promoted the concept of credibility, believing that the credibility of all evidence is equal, and modified the Dempster rule by weighted summation [17]. Yang established a unique evidential reasoning (ER) rule, combining multiple pieces of independent evidence with weight and reliability, improving and enhancing the Dempster rule by determining how to combine completely reliable evidence and analyzing significant or complete conflicts through new reliability disturbances [18]. Deng Yong proposed a new concept called generalized evidence theory (GET), and defined a new concept called generalized basic probability assignment (GBPA). They established a model to deal with uncertain information, provided generalized combination rules (GCR) for the combination of GBPA, and constructed a generalized conflict model to measure the conflict between evidence [19]. The second aspect is to modify the original evidence. Smets proposed pignistic probability transformations and adopted an average distribution strategy in order for the mass function of assignments to meet the conditions [20]. Based on a geometric interpretation of this evidence theory, Jousselme defined the Jousselme evidence distance to describe the differences in evidence through distance information [21]. Murphy believed that the evidence should be a weighted average, which would better deal with the normalization problem [22]. Tang proposed a new multi-sensor data fusion method based on weighted confidence entropy, which measures the uncertainty of evidence through mass functions and an identification framework to reduce the loss of evidence information [23].

In this field, domestic and foreign scholars and experts have achieved excellent results in a particular range. However, in the application of single target recognition, there is only a single result, i.e., $m(A_{1})$. Multi-subset BPAs such as $m(A_{1}A_{2})$, which means there is still uncertainty in the outcome of the fusion, reduce the probability of target recognition. At present, there has been no relevant research conducted on the matter. In this article, a method for recognizing single targets through conflicting evidence fusion is proposed. This method involves consolidating multiple subsets within the framework of evidence theory into a single subset, while incorporating the evidence distance, evidence angle, and entropy to enhance the accuracy of target fusion. First, the pignistic probability function is improved to transform each original evidence into a single propositional subset to avoid a single recognition result including multiple subset propositions in the process of the Dempster rule. Then, the combination of the Manhattan distance and evidence angle measurements is proposed to extract the degree of evidence certainty and obtain the mutual support information between all data. Furthermore, entropy is introduced to calculate the uncertainty of evidence. The initial evidence is fused according to the coefficient of uncertainty and is transformed to updated evidence. Finally, the Dempster combination rule is used to fuse the updated evidence.

## 2. Materials and Methods

### 2.1. Dempster Rule

Let *U* be a domain set representing all values of *X*, and all elements in *U* are not integrated. Then, *U* is called the recognition framework of *X*.

The research objects in scientific theory compose a nonempty set, which is called a domain.

**Definition 1**.*Let U be a recognition framework, then the function* $m:2^{U}\to \left[0,1\right]$ *satisfies the following conditions:**(1)* m(∅)=0*;**(2)* $\sum_{A\subset U}=1$*.*

Then, m(A) is the basic probability assignment (BPA) of *A*, m(A) is the mass function, and m(A) represents the degree of trust in A. If m(A)>0, *A* is called a focal element.

Hypothesis $m_{1}$ and $m_{2}$ are the two basic probability assignments on the same recognition framework *U*, and the focal elements are $A_{1},A_{2},\ldots ,A_{k}$ and $B_{1},B_{2},\ldots ,B_{r}$. Namely,
(1)$$K=\sum_{A_{i}\cap B_{j}=\emptyset }m_{1}\left(A_{i}\right)m_{2}\left(B_{j}\right)<1$$

Then,
(2)$$m\left(C\right)=\left\{\begin{array}{ccc} \frac{\sum_{A_{i}\cap B_{j}=\emptyset }m_{1}\left(A_{i}\right)m_{2}\left(B_{j}\right)}{1-K} & , & \forall C\subset U\\ 0 & , & C=\emptyset . \end{array}\right.$$
where i=1,2,…,k; j=1,2,…,r; and *K* is the conflict factor, which reflects the degree of conflict between evidence. $\frac{1}{1-K}$ is called the normalization factor. The Dempster rule allocates the conflict to each set in equal proportion.

Define the system identification framework as $U=\{A_{1},A_{2},\ldots ,A_{M}\}$, *N* evidence as $E_{1},E_{2},\ldots ,E_{N}$, and the mass functions corresponding to each evidence as $m_{1},m_{2},\ldots ,m_{N}$.

### 2.2. Improved Pignistic Probability Function

In single target recognition, the fusion recognition result often only has a certain target. In the framework of DS evidence theory, when evidence contains multiple subset propositions, the fusion result also contains multiple subset propositions, which increases the computational complexity. This work improves the pignistic probability function to transform the multiple subset propositions into single subset propositions, in which the BPA of multiple subset propositions is distributed according to the weight of single subset propositions. The weight is allocated according to the information of each single subset proposition provided by the evidence itself, which reduces the computational complexity and information lost in the process of transforming the pignistic probability function [24] from the BPA to the single subset BPA. The improved pignistic probability conversion function is as follows:(3)$$BetP_{m}\left(A_{i}\right)=\sum_{A_{i}\cup A_{j}\subseteq U,B\subseteq A_{j}}\frac{m(A_{i})}{\sum_{k=1}^{|A_{j}|}m\left(B\right)}\cdot \frac{m(A_{j})}{1-m(\phi )}$$
where *A* is the proposition of original evidence and *B* is a simple subset proposition in $A_{j}$, $A_{i},A_{j}$ refers to the multi sub proposition in evidence, $A_{i}\neq A_{j}$, ∅ is an empty set, m(∅)≠1, and $|A_{j}|$ represents the number of elements contained in proposition *A*.

After the pignistic probability function is conserved, the BPA is converted into a single subset BPA $m_{1}^{\prime },m_{2}^{\prime },\ldots ,m_{N}^{\prime }$.
(4)$$m_{i}^{\prime }=\left\{BetP_{m_{i}}\left(A_{1}\right),BetP_{m_{i}}\left(A_{2}\right),\ldots ,BetP_{m_{i}}\left(A_{N}\right)\right\}$$

### 2.3. Evidence Support Based on the Manhattan Distance

The distance between evidence [25] can effectively measure the degree of support between evidence. At present, domestic and foreign scholars have proposed a variety of methods to measure distance, including the Lance distance, the Jousselme distance, and the Mahalanobis distance. However, the Lance distance does not take into account the correlation between indicators. The Jousselme distance is affected by the dispersion of the basic probability distribution of evidence [26]. The Mahalanobis distance function requires calculation of the covariance of the matrix, which is hugely complex. The Manhattan distance introduced in this paper calculates the distance of each single subset BPA identification result to measure the similarity between the evidence. This method has low computational complexity.

The Manhattan distance between two pieces of evidence is calculated as: (5)$$d\left(E_{i},E_{j}\right)=\sum_{k=1}^{M}|BetP_{m_{i}}\left(A_{k}\right)-BetP_{m_{j}}\left(A_{k}\right)|$$
where i,j=1,2,…,N. The Manhattan distance between each evidence is calculated to obtain the distance matrix D: (6)$$D=\left[\begin{array}{cccc} d(E_{1},E_{1}) & d(E_{2},E_{1}) & \cdots & d(E_{N},E_{1})\\ d(E_{1},E_{2}) & d(E_{2},E_{2}) & \cdots & d(E_{N},E_{2})\\ \cdots & \cdots & \cdots & \ldots \\ d(E_{1},E_{N}) & d(E_{2},E_{N}) & \cdots & d(E_{N},E_{N}) \end{array}\right]$$

The distance between evidence is negatively correlated with the support.

The calculation of the evidence support of $E_{1},E_{2},\ldots ,E_{N}$ is: (7)$$SUP_{i}=\frac{1}{N-1}\sum_{k=1,k\neq i}^{N}\frac{1}{D_{i}k}$$

The support degree is obtained based on the Manhattan distance between evidence. The support is normalized to obtain the support coefficient of the evidence $Cor\text{\_}d(E_{i})$.
(8)$$Cor\text{\_}d\left(E_{i}\right)=\frac{SUP_{i}}{\sum_{j=1}^{N}SUP_{j}}$$

### 2.4. Evidence Similarity Based on Evidence Angle

The angle between the two pieces of evidence can be used to structure the consistency between the evidence subjects, and the results obtained can be used to measure the similarity between the two evidence subjects. The formula for the evidence angle [27] is as follows:(9)$$cos\left(E_{i},E_{j}\right)=\frac{m_{i}^{\prime }\times m_{j}^{\prime }}{∥m_{i}^{\prime }∥\times ∥m_{j}^{\prime }∥}=\frac{\sum_{k=1}^{M}BetP_{m_{i}}\left(A_{k}\right)\times BetP_{m_{j}}\left(A_{k}\right)}{\sqrt{\sum_{k=1}^{M}{\left[BetP_{m_{i}}\left(A_{k}\right)\right]}^{2}}\times \sqrt{\sum_{k=1}^{M}{\left[BetP_{m_{j}}\left(A_{k}\right)\right]}^{2}}}$$

The larger value of $cos(E_{i},E_{j})$, the more consistent two pieces of evidence are. It shows that there is a higher similarity between the two pieces of evidence. The evidence angle between each evidence is calculated from the angle matrix, Ang. The similarity between evidence is calculated from the angle matrix:(10)$$SIM_{i}=\frac{1}{N-1}\sum_{k=1,k\neq i}^{N}Ang_{ik}$$

The similarity between evidence is measured based on the evidence angle. The similarity is normalized to obtain the similarity coefficient of the evidence $Cor_{A}\left(E_{i}\right)$.
(11)$$Cor\text{\_}A\left(E_{i}\right)=\frac{SIM_{i}}{\sum_{j=1}^{N}SIM_{j}}$$

### 2.5. Evidence Uncertainty Based on Entropy

In evidence theory, the amount of information content that evidence carries can be measured by information entropy. The higher the information entropy, the more information the evidence carries, and the lower the probability of occurrence in the real world, the higher the uncertainty. The calculation formula for information entropy is as follows:(12)$$H\left(E_{i}\right)=-\sum_{A_{k}\subset E_{i}}m_{i}\left(A_{k}\right)log_{2}\left(m_{i}\left(A_{k}\right)\right)$$

The uncertainty coefficient of each piece of evidence $Cor\text{\_}S(E_{i})$ is calculated by: (13)$$Cor\text{\_}S\left(E_{i}\right)=e^{H_{n}\left(E_{i}\right)}$$
where i=1,2,…,N.

### 2.6. Evidence Fusion Based on the Dempster Rule

The evidence fusion coefficient integrating the Manhattan distance, the evidence angle, and the reliability entropy is: (14)$$Cor\left(E_{i}\right)=Cor\text{\_}S\left(E_{i}\right)\times Cor\text{\_}d\left(E_{i}\right)\times Cor\text{\_}A\left(E_{i}\right)$$

The fusion coefficient $Cor(E_{i})$ is normalized to obtain the final evidence fusion coefficient $Cor\text{\_}fusion(E_{i})$. The single subset BPA $\left\{m_{1}^{\prime },m_{2}^{\prime },\ldots ,m_{N}^{\prime }\right\}$ is modified:(15)$$m^{\prime \prime }=\sum_{i=1}^{N}Cor_{f}usion\left(E_{i}\right)\times m_{i}^{\prime }$$
where i=1,2,…,N;

All the initial evidence is replaced with $m^{\prime \prime }$, and finally the modified evidence is fused with the Dempster rules: (16)$$m_{f}us={\left({\left({\left(m^{\prime \prime }⊕m^{\prime \prime }\right)}_{1}\cdots \right)}_{i}⊕m^{\prime \prime }\right)}_{N-1}$$

The flow graph of the method proposed in this paper is shown in Figure 1.

## 3. Results

In the following, we verify the effectiveness of the method through two conflicting examples: those that only contain single-subset propositions and those that contain multiple-subset propositions.

### 3.1. An Example of Single-Subset Proposition Conflicting Evidence

An example of single-subset proposition conflicting evidence can be found in reference [28]. An evidence recognition framework is assumed and there are five independent pieces of evidence. The corresponding BPA is shown in Table 1 [28].

#### 3.1.1. Improved Pignistic Probability Function

This example is a simple subset proposition, and the conversed BPA is obtained by Formula (1). $m_{1}^{\prime }=\left\{0.90,0,0.10\right\},m_{2}^{\prime }=\left\{0,0.01,0.99\right\},m_{3}^{\prime }=\left\{0.50,0.20,0.30\right\},m_{4}^{\prime }=\left\{0.98,0.01,0.01\right\},m_{5}^{\prime }=\left\{0.90,0.05,0.05\right\}$.

#### 3.1.2. Calculate Fusion Coefficient

Apply Formulas (5)–(8) as follows to obtain the evidence support coefficient, the distance matrix, and support matrix as: $$D=\left[\begin{array}{ccccc} 0 & 1.8 & 0.8 & 0.18 & 0.10\\ 1.80 & 0 & 1.38 & 1.96 & 1.88\\ 0.80 & 1.38 & 0 & 0.96 & 0.80\\ 0.18 & 1.96 & 0.96 & 0 & 0.16\\ 0.10 & 1.88 & 0.80 & 0.16 & 0 \end{array}\right]$$

The support based on the Manhattan distance between evidence is: $$SUP_{i}=\left\{4.3425,0.5800,1.0650,3.3400,4.5075\right\}$$

The evidence support coefficient $Cor(E_{i})$ is: $$Cord\text{\_}d\left(E_{i}\right)=\left\{0.3139,0.0419,0.0770,0.2414,0.3258\right\}$$

Apply Formulas (9)–(11) as follows to obtain the the evidence similarity coefficient $Cor\text{\_}A(E_{i})$,
$$\begin{array}{c} Ang=\left[\begin{array}{cccccc} 1 & 0.11 & 0.86 & 0.99 & 1.00 & \\ 0.11 & 1 & 0.49 & 0.01 & 0.06\\ 0.86 & 0.49 & 1 & 0.82 & 0.85\\ 0.99 & 0.01 & 0.82 & 1 & 1.00\\ 1.00 & 0.06 & 0.85 & 1.00 & 1 \end{array}\right]\\ Ang_{i}=\left\{0.7400,0.1675,0.7550,0.7050,0.7275\right\} \end{array}$$

The evidence similarity coefficient $Cor\text{\_}A(E_{i})$ is: $$Cord\text{\_}A\left(E_{i}\right)=\left\{0.2391,0.0541,0.2439,0.2278,0.2351\right\}$$

Apply Formulas (12) and (13) as follows to obtain the evidence uncertainty coefficient $Cor\text{\_}S(E_{i})$.
$$Cord\text{\_}S\left(E_{i}\right)=\left\{1.60,1.08,4.44,1.17,1.77\right\}$$

The support coefficient and similarity coefficient represent the certainty of evidence, and the uncertainty coefficient represents the uncertainty of evidence. The final evidence fusion coefficient is obtained by applying Formulas (14) and (15): $$Cord\text{\_}fusion\left(E_{i}\right)=\left\{0.2960,0.0059,0.2055,0.1585,0.3342\right\}$$

Modify the evidence again
$$m^{\prime \prime }=\sum_{i=1}^{N}Cor\text{\_}fusion\left(E_{i}\right)\times m_{i}^{\prime }=\left\{0.8253,0.0595,0.1154\right\}$$
to replace the initial evidence.

#### 3.1.3. Evidence Fusion Based on the Dempster Rule

Formula (16) is applied for fusion four times, and the fusion results are shown in Table 2 and Figure 2. A comparison with other methods is shown in Table 3 and Figure 3.

### 3.2. An Example of Multi-Subset Proposition Conflicting Evidence

Suppose there is a multi-sensor-based target recognition system, then the recognized targets are $U=\{A_{1},A_{2},A_{3}\}$, which are the real targets. There are five independent sensors. The recognition results of the five sensors are shown in Table 4.

#### 3.2.1. Improved Pignistic Probability Function

According to Formula (3), the conversed BPA is as shown in Table 5.

#### 3.2.2. Calculate Fusion Coefficient $Cor(E_{i})$

As calculated by Formulas (4)–(15), the coefficients are shown in Table 6.

We obtained the final evidence of the BPA: $$m^{\prime \prime }=\sum_{i=1}^{N}Cord\text{\_}fusion\left(E_{i}\right)*m_{i}^{\prime }=\left\{0.8284,0.0873,0.0841\right\}$$

#### 3.2.3. Evidence Fusion Based on the Dempster Rule

Evidence fusion was performed four times by the Dempster rule, and the fusion results are shown in Table 7 and Figure 4. A comparison with other methods is shown in Table 8 and Figure 5.

## 4. Discussion

As shown in the figure above, applying the Dempster fusion rule leads to counter-intuitive results.

The fusion results of single- and multi-subset conflicting evidence are discussed in this section. Analyzing Table 2 and Table 7 and Figure 2 and Figure 4, our proposed method has a good fusion effect, and the BPA reaches 0.9999 and 1.0000 in the fourth fusion. The BPA decreases with the increase in fusion time. It shows that the method proposed in this paper can effectively extract the characteristics of the evidence.

Analyzing Table 3 and Table 8 and Figure 3 and Figure 5, when the number of pieces of evidence is two or three, our method is not as effective as Chen’s method and Zhao’s method. In the case of a small amount of evidence, the input data are insufficient and it is difficult to extract multiple features from each evidence source. However, with the increase in the number of evidence sources, our method’s accuracy rapidly improves and its accuracy performance is expected to be even better. According to the tables and the figures, it can be seen that our proposed method has a higher accuracy and better effect after three or more fusion processes. Furthermore, for four or more fusion processes, our proposed method has a higher accuracy and better impact in the fusion results of multi-subset conflict examples.

Experiments have shown that the method proposed in this article can effectively extract mutually supportive features between various evidence sources when there are sufficient evidence sources and can achieve good results.

## 5. Conclusions

In this article, we proposed a novel evidence combination method based on an improved pignistic probability function. Considering evidence characteristics and information richness, this paper proposes a novel method to solve the problem of highly conflicting evidence fusion in DS evidence theory. Through experiments, it has been shown that we have achieved good results in dealing with single target recognition problems and an improved fusion accuracy of the evidence theory framework in target fusion recognition. Evidence theory has a strong ability to handle uncertainty problems. Our next work will further investigate how evidence theory can be extended and applied in the real world.

## Figures and Tables

**Figure 1 entropy-25-00948-f001:**
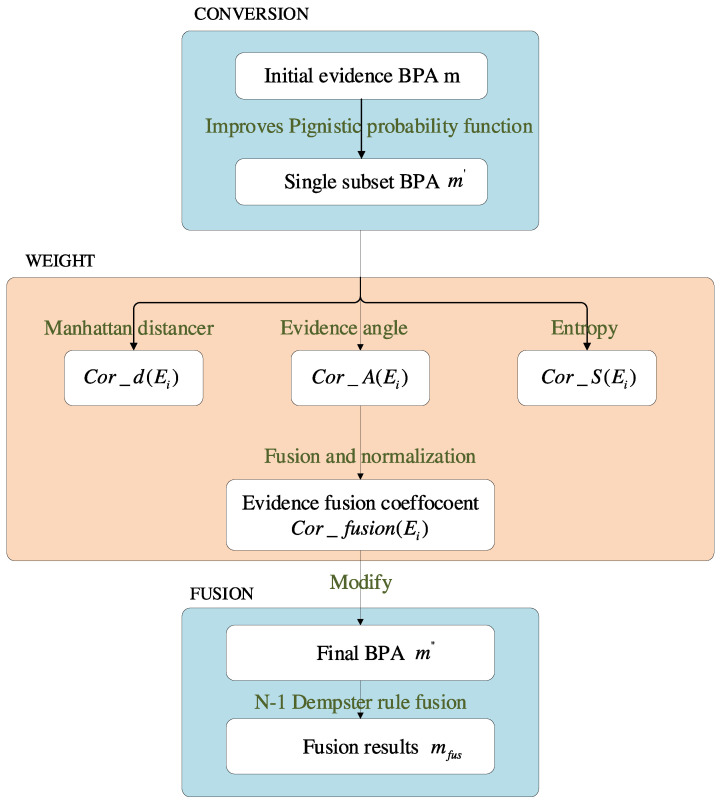
The flow graph of the proposed method.

**Figure 2 entropy-25-00948-f002:**
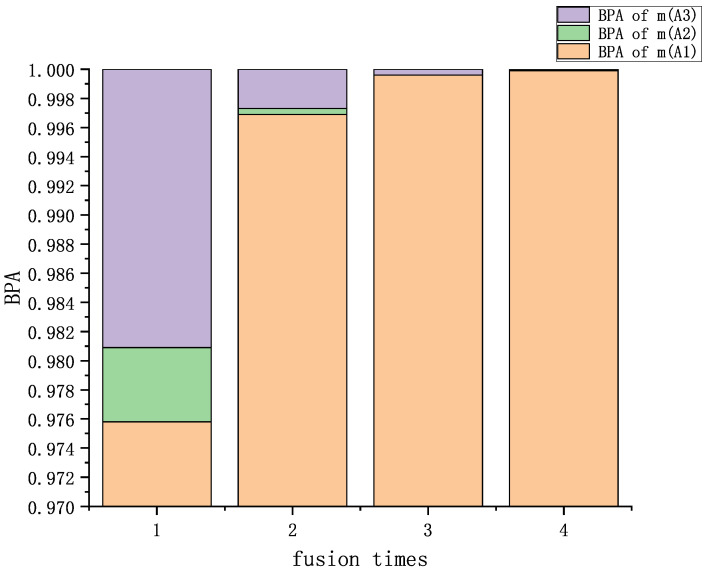
Fusion results of multi-subset proposition conflicting examples.

**Figure 3 entropy-25-00948-f003:**
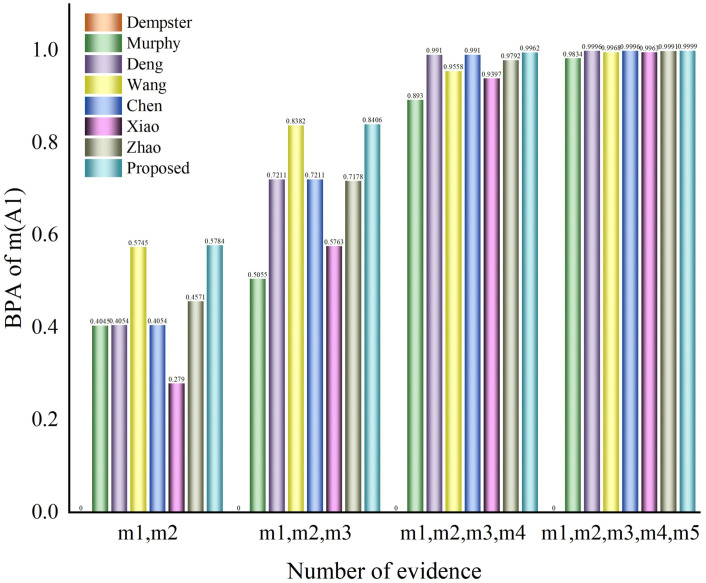
Fusion results chart of a comparison of different methods of fusion of several pieces of single-subset proposition evidence.

**Figure 4 entropy-25-00948-f004:**
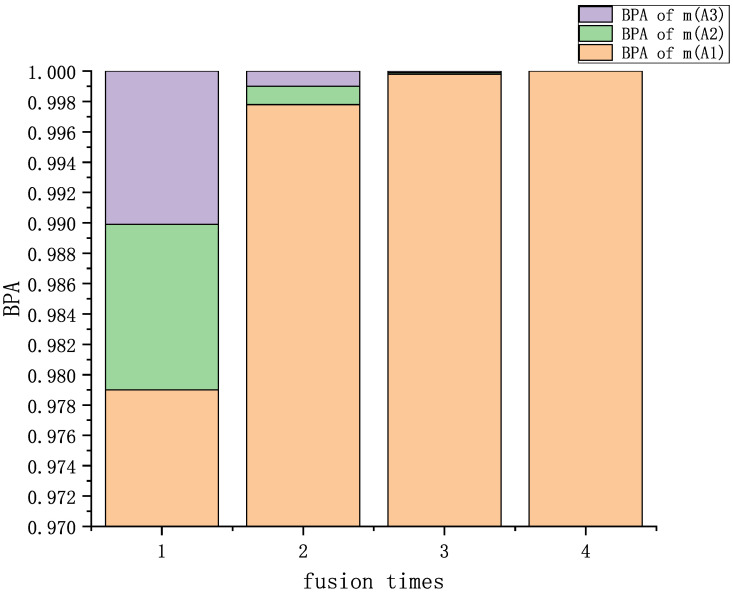
Fusion results of multi-subset proposition conflict.

**Figure 5 entropy-25-00948-f005:**
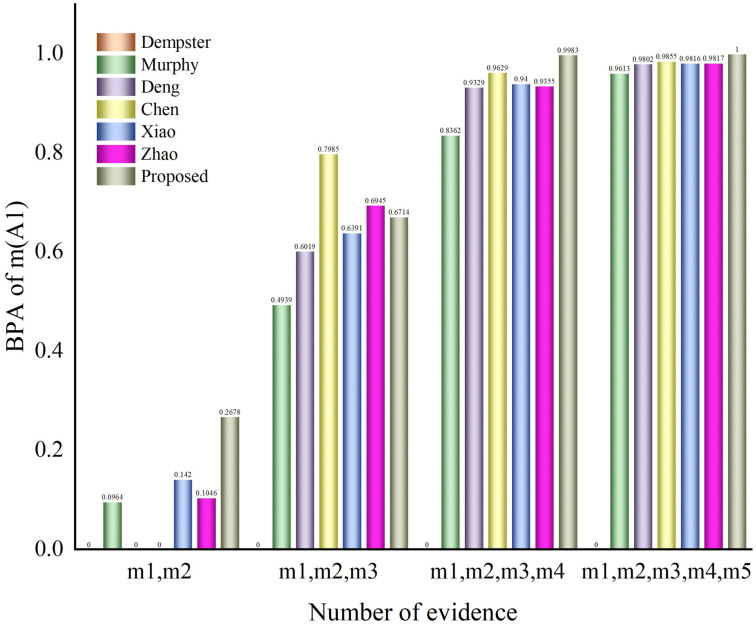
Fusion results chart of a comparison of different methods on the fusion of several pieces of multi-subset proposition evidence.

**Table 1 entropy-25-00948-t001:** Single-subset proposition conflicting evidence.

Evidence	$m(A_{1})$	$m(A_{2})$	$m(A_{3})$
$E_{1}$	0.90	0	0.10
$E_{2}$	0	0.01	0.99
$E_{3}$	0.50	0.20	0.30
$E_{4}$	0.98	0.01	0.01
$E_{5}$	0.90	0.05	0.05

**Table 2 entropy-25-00948-t002:** Fusion results of single-subset proposition conflicting examples.

Fusion Times	$m(A_{1})$	$m(A_{2})$	$m(A_{3})$
First	0.9758	0.0051	0.0191
Second	0.9969	0.0004	0.0027
Third	0.9996	0.0000	0.0004
Fourth	0.9999	0.0000	0.0001

**Table 3 entropy-25-00948-t003:** Comparison of evidence fusion with different methods.

Approach	Fusion Result				
	BPA	$m\left(A_{1}\right),m\left(A_{2}\right)$	$m\left(A_{1}\right),m\left(A_{2}\right),$ $m(A_{3})$	$m\left(A_{1}\right),m\left(A_{2}\right),$ $m\left(A_{3}\right),m\left(A_{4}\right)$	$m\left(A_{1}\right),m\left(A_{2}\right),m\left(A_{3}\right),$ $m\left(A_{4}\right),m\left(A_{5}\right)$
	$m(A_{1})$	0	0	0	0
Dempster-Shafer	$m(A_{2})$	0	0	0	0
	$m(A_{3})$	1	1	1	1
	$m(A_{1})$	0.4054	0.5055	0.8930	0.9834
Murphy [22]	$m(A_{2})$	0.0001	0.0000	0.0001	0.0000
	$m(A_{3})$	0.5946	0.4945	0.1069	0.0166
	$m(A_{1})$	0.4054	0.7211	0.9910	0.9996
Deng [29]	$m(A_{2})$	0.0001	0.0040	0.0001	0.0000
	$m(A_{3})$	0.5946	0.2749	0.0089	0.0003
	$m(A_{1})$	0.5745	0.8382	0.9558	0.9968
Wang [28]	$m(A_{2})$	0.0033	0.0142	0.0010	0.0001
	$m(A_{3})$	0.4223	0.1476	0.0431	0.0031
	$m(A_{1})$	0.4054	0.7211	0.9910	0.9996
Chen [30]	$m(A_{2})$	0.0001	0.0040	0.0001	0.0000
	$m(A_{3})$	0.5946	0.2749	0.0089	0.0003
	$m(A_{1})$	0.2790	0.5763	0.9397	0.9963
Xiao [31]	$m(A_{2})$	0.0001	0.0065	0.0004	0.0000
	$m(A_{3})$	0.7210	0.4173	0.0599	0.0037
	$m(A_{1})$	0.4571	0.7178	0.9792	0.9991
Zhao [32]	$m(A_{2})$	0.0000	0.0046	0.0001	0.0000
	$m(A_{3})$	0.5429	0.2775	0.0207	0.0009
	$m(A_{1})$	0.5784	0.8406	0.9962	0.9999
Ours	$m(A_{2})$	0.0000	0.0187	0.0002	0.0000
	$m(A_{3})$	0.4216	0.1407	0.0036	0.0001

**Table 4 entropy-25-00948-t004:** Single-subset proposition conflicting evidence.

Evidence	$m(A_{1})$	$m(A_{2})$	$m(A_{3})$	$m(A_{1}A_{3})$
$E_{1}$	0.41	0.29	0.30	0.00
$E_{2}$	0.00	0.90	0.10	0.00
$E_{3}$	0.58	0.07	0.00	0.35
$E_{4}$	0.55	0.10	0.00	0.35
$E_{5}$	0.60	0.00	0.10	0.30

**Table 5 entropy-25-00948-t005:** Conversed BPA.

Evidence	$m(A_{1})$	$m(A_{2})$	$m(A_{3})$
$E_{1}$	0.41	0.29	0.30
$E_{2}$	0.00	0.90	0.10
$E_{3}$	0.93	0.07	0.00
$E_{4}$	0.90	0.10	0.00
$E_{5}$	0.8571	0.00	0.1429

**Table 6 entropy-25-00948-t006:** Fusion coefficients of multi-subset proposition evidence.

Coefficient	$E_{1}$	$E_{2}$	$E_{3}$	$E_{4}$	$E_{5}$
$Cor_{d}\left(E_{i}\right)$	0.0440	0.0221	0.2924	0.3212	0.3203
$Cor_{A}\left(E_{i}\right)$	0.2352	0.0629	0.2336	0.2376	0.2307
$Cor_{S}\left(E_{i}\right)$	4.7894	1.5984	1.4470	1.5984	1.8072
$Cor(E_{i})$	0.1221	0.0054	0.2433	0.3004	0.3287

**Table 7 entropy-25-00948-t007:** Fusion results of multi-subset proposition conflict examples.

Fusion Times	$m(A_{1})$	$m(A_{2})$	$m(A_{3})$
First	0.9790	0.0109	0.0101
Second	0.9978	0.0012	0.0010
Third	0.9998	0.0001	0.0001
Fourth	1.0000	0.0000	0.0000

**Table 8 entropy-25-00948-t008:** Comparison of evidence fusion with different methods.

Approach	Fusion Result				
	BPA	$m\left(A_{1}\right),m\left(A_{2}\right)$	$m\left(A_{1}\right),m\left(A_{2}\right),$ $m(A_{3})$	$m\left(A_{1}\right),m\left(A_{2}\right),$ $m\left(A_{3}\right),m\left(A_{4}\right)$	$m\left(A_{1}\right),m\left(A_{2}\right),m\left(A_{3}\right),$ $m\left(A_{4}\right),m\left(A_{5}\right)$
	$m(A_{1})$	0	0	0	0
Dempster-Shafer	$m(A_{2})$	0.8969	0.6350	0.3320	0
	$m(A_{3})$	0.1031	0.3650	0.6680	1
	$m(A_{1})$	0.0964	0.4939	0.8362	0.9613
Murphy [22]	$m(A_{2})$	0.8119	0.4180	0.1147	0.0147
	$m(A_{3})$	0.0917	0.0792	0.0410	0.0166
	$m(A_{1}A_{3})$	0.0000	0.0090	0.0081	0.0032
	$m(A_{1})$	0.0000	0.6019	0.9329	0.9802
Deng [29]	$m(A_{2})$	0.8969	0.2908	0.0225	0.0009
	$m(A_{3})$	0.1031	0.0991	0.0354	0.0154
	$m(A_{1}A_{3})$	0.0000	0.0082	0.0092	0.0035
	$m(A_{1})$	0.0000	0.7985	0.9629	0.9855
Chen [30]	$m(A_{2})$	0.8969	0.1060	0.0043	0.0001
	$m(A_{3})$	0.1031	0.0752	0.0190	0.0096
	$m(A_{1}A_{3})$	0.0000	0.0203	0.0139	0.0048
	$m(A_{1})$	0.1420	0.6391	0.9400	0.9816
Xiao [31]	$m(A_{2})$	0.7412	0.2462	0.0165	0.0006
	$m(A_{3})$	0.1168	0.1072	0.0341	0.0141
	$m(A_{1}A_{3})$	0.0000	0.0075	0.0093	0.0037
	$m(A_{1})$	0.1046	0.6945	0.9355	0.9817
Zhao [32]	$m(A_{2})$	0.7989	0.1902	0.0163	0.0000
	$m(A_{3})$	0.0965	0.1062	0.0409	0.0147
	$m(A_{1}A_{3})$	0.0000	0.0091	0.0073	0.0036
	$m(A_{1})$	0.2678	0.6714	0.9983	1.0000
Ours	$m(A_{2})$	0.5551	0.2205	0.0015	0.0000
	$m(A_{3})$	0.1771	0.1080	0.0001	0.0000

## Data Availability

No new data were created or analyzed in this study. Data sharing is not applicable to this article.

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
