# Peer review of "A Novel Evidence Combination Method Based on Improved Pignistic Probability"

_entropy, 2023, doi:10.3390/e25060948_

Round 1
Reviewer 1 Report
The motivations of the research are not presented in the paper . Why the bayesian probability approach is not appropriated to update and to combine new evidences? Which are the the problem that the method proposed in the paper should solve? No theorethical aspects are are investigated sufficiently in the paper.
Sufficient quality of Engish language
Author Response
Point 1: The motivations of the research are not presented in the paper .
Response 1: Thank you for your comments.The motivations of the research is to solve the problem of conflicting evidence fusion in single target recognition. In the application of single target recognition there is only a single result, like m(A1). Multi subset BPA like m(A1,A2), which means there is still uncertainty in the outcome of the fusion, reduced the probability of target recognition.
We have made modifications to the abstract and introduction sections of the article, and the modified sections have been marked.
Point 2: Why the bayesian probability approach is not appropriated to update and to combine new evidences?
Response 2: Firstly, the probability in Bayesian cannot distinguish between ignorance and equipossibility, but rather regards ignorance as equipossibility. And the evidence theory can distinguish between using to indicate ignorance, and using m (a)=m (b) to indicate other possibilities. Secondly, if the probability of believing in proposition A is S, then the degree of belief in the inverse of proposition A is: 1-S. And using the definition of the basic probability assignment function in evidence theory, there is . Thirdly, the probability function is a single valued function, while the trust function is a set variable function, which can more easily express "rough" information.
Point 3: Which are the the problem that the method proposed in the paper should solve?
Response 3: Thank you for your comments. To solve the problem of conflicting evidence fusion in single target recognition. Improving the fusion recognition accuracy of evidence theory in single target recognition is also a new evidence theory conflict evidence fusion method.We have made modifications to the abstract and introduction sections of the article, and the modified sections have been marked.
Point 4: No theorethical aspects are are investigated sufficiently in the paper.
Response 4: In response to the problem of single target recognition fusion, we propose a new conflict evidence fusion method within the framework of evidence theory. By combining single target recognition with conflict evidence problems, we transform multiple subsets into single subsets and extract fusion features between multiple evidences. Compared with classic and recent methods, experiments show that our method performs well. We have made some improvements to the initial draft and hope that you can continue to provide valuable comments on our article.We updated the manuscript.

Reviewer 2 Report
The article discusses evidence theory as a method for uncertain reasoning and its ability to express uncertainty and ignorance directly. While evidence theory is more intuitive and easier to obtain primary data than probability reasoning theory, it often accounts for counterintuitive results when faced with conflicting evidence, resulting in the "Zadeh paradox." In recent literature, there have been studies that improved evidence theory by modifying the fusion rules and original evidence. The paper proposes a method suitable for the fusion of single subset propositional evidence and multi subset propositional evidence and improves the Pignistic probability function to transform each original evidence into a single subset propositional. Additionally, it introduces the combination of Manhattan distance and evidence angle measurement to extract the degree of evidence certainty and mutual support information between all evidence. Finally, the updated evidence is fused using the Dempster combination rule.
The article provides a clear and concise explanation of the background and motivation for the proposed research, as well as a good overview of the proposed method and how it improves on existing approaches. For these reasons, I am in favor of it being published in the journal.
Author Response
Point 1: The article discusses evidence theory as a method for uncertain reasoning and its ability to express uncertainty and ignorance directly. While evidence theory is more intuitive and easier to obtain primary data than probability reasoning theory, it often accounts for counterintuitive results when faced with conflicting evidence, resulting in the "Zadeh paradox." In recent literature, there have been studies that improved evidence theory by modifying the fusion rules and original evidence. The paper proposes a method suitable for the fusion of single subset propositional evidence and multi subset propositional evidence and improves the Pignistic probability function to transform each original evidence into a single subset propositional. Additionally, it introduces the combination of Manhattan distance and evidence angle measurement to extract the degree of evidence certainty and mutual support information between all evidence. Finally, the updated evidence is fused using the Dempster combination rule.
The article provides a clear and concise explanation of the background and motivation for the proposed research, as well as a good overview of the proposed method and how it improves on existing approaches. For these reasons, I am in favor of it being published in the journal.
Response 1: Thank you very much for your recognition. We will continue to work hard and continue our research on evidence theory.

Reviewer 3 Report
The paper is devoted to evidence theory which has some advantages over classical Bayesian probability theory. It is connected with a novel combination method connected with improved Pignistic probability. They introduce a new probability function based on using weight of single subset proposition in basic probability assignment. Authors use Manhattan distance and evidence angle measurement. They obtain BPA for different values of parameters and compare them with another approaches.
One of the main problems is connected with the scientific soundness of the work. Principal results are quite close to the ones described in previous works in this field, and the method described Jousselme et al. works better (it even can be seen in the tables and figures). It is quite strange because a lot of of such works were written in the early 2000s. The authors should strongly explain why it is necessary to present their results.
Also there are some stylistic problems connected with presentation of the text.
There are a lot of spelling mistakes. Also the authors often finish the sentences using ";" which is not right.
Author Response
Point 1: The paper is devoted to evidence theory which has some advantages over classical Bayesian probability theory. It is connected with a novel combination method connected with improved Pignistic probability. They introduce a new probability function based on using weight of single subset proposition in basic probability assignment. Authors use Manhattan distance and evidence angle measurement. They obtain BPA for different values of parameters and compare them with another approaches.
One of the main problems is connected with the scientific soundness of the work. Principal results are quite close to the ones described in previous works in this field, and the method described Jousselme et al. works better (it even can be seen in the tables and figures). It is quite strange because a lot of of such works were written in the early 2000s. The authors should strongly explain why it is necessary to present their results.
Response 1: Thank you for your comments. I'm sorry that some of my mistakes in my work caused me to make a mistake in the writing. The Jousselme distance in the article is a method used by Deng in 2004. Through experimental tables and images, we found that Chen and Zhao's method performed better than ours when the number of evidence was only two or three. However, as the number of evidence increased to four or five, our method's accuracy improved rapidly and exceeded some classic methods in the experiment and new methods in recent years. The first three comparative methods are classic old methods in evidence theory, while the last three are new methods in recent years.We updated the manuscript by modified discussion and conclusion, and the modified sections have been marked.
Point 2: Also there are some stylistic problems connected with presentation of the text. There are a lot of spelling mistakes. Also the authors often finish the sentences using ";" which is not right.
Response 2: Thank you for your comments. We corrected the syntax error in my paper. For ';', We have also made modifications.We updated the manuscript and the modified sections have been marked.

Round 2
Reviewer 1 Report
no comments
Reviewer 3 Report
Dear Authors!
Thank you for your response. To my mind, some important changes have been made. However, it could be important to give some more links, but the minimum requirements have been met.
Best regards,
Reviewer